# Clinical Severity of SARS-CoV-2 Omicron Variant Compared with Delta among Hospitalized COVID-19 Patients in Belgium during Autumn and Winter Season 2021–2022

**DOI:** 10.3390/v14061297

**Published:** 2022-06-14

**Authors:** Nina Van Goethem, Pui Yan Jenny Chung, Marjan Meurisse, Mathil Vandromme, Laurane De Mot, Ruben Brondeel, Veerle Stouten, Sofieke Klamer, Lize Cuypers, Toon Braeye, Lucy Catteau, Louis Nevejan, Joris A. F. van Loenhout, Koen Blot

**Affiliations:** 1Scientific Directorate of Epidemiology and Public Health, Sciensano, J. Wytsmanstraat 14, 1050 Brussels, Belgium; jenny.chung@sciensano.be (P.Y.J.C.); marjan.meurisse@sciensano.be (M.M.); mathil.vandromme@sciensano.be (M.V.); laurane.demot@sciensano.be (L.D.M.); ruben.brondeel@sciensano.be (R.B.); veerle.stouten@sciensano.be (V.S.); sofieke.klamer@sciensano.be (S.K.); toon.braeye@sciensano.be (T.B.); lucy.catteau@sciensano.be (L.C.); joris.vanloenhout@sciensano.be (J.A.F.v.L.); koen.blot@sciensano.be (K.B.); 2Clinical Department of Laboratory Medicine and National Reference Center for Respiratory Pathogens, University Hospitals Leuven, 3000 Leuven, Belgium; lize.cuypers@uzleuven.be (L.C.); louis.nevejan@uzleuven.be (L.N.)

**Keywords:** SARS-CoV-2, COVID-19, Omicron, Delta, genomic surveillance

## Abstract

This retrospective multi-center matched cohort study assessed the risk for severe COVID-19 (combination of severity indicators), intensive care unit (ICU) admission, and in-hospital mortality in hospitalized patients when infected with the Omicron variant compared to when infected with the Delta variant. The study is based on a causal framework using individually-linked data from national COVID-19 registries. The study population consisted of 954 COVID-19 patients (of which, 445 were infected with Omicron) above 18 years old admitted to a Belgian hospital during the autumn and winter season 2021–2022, and with available viral genomic data. Patients were matched based on the hospital, whereas other possible confounders (demographics, comorbidities, vaccination status, socio-economic status, and ICU occupancy) were adjusted for by using a multivariable logistic regression analysis. The estimated standardized risk for severe COVID-19 and ICU admission in hospitalized patients was significantly lower (RR = 0.63; 95% CI (0.30; 0.97) and RR = 0.56; 95% CI (0.14; 0.99), respectively) when infected with the Omicron variant, whereas in-hospital mortality was not significantly different according to the SARS-CoV-2 variant (RR = 0.78, 95% CI (0.28–1.29)). This study demonstrates the added value of integrated genomic and clinical surveillance to recognize the multifactorial nature of COVID-19 pathogenesis.

## 1. Introduction

The clinical spectrum of coronavirus disease 19 (COVID-19), resulting from infection with the severe acute respiratory syndrome coronavirus 2 (SARS-CoV-2), ranges from asymptomatic infections or mild respiratory symptoms to severe pneumonia and acute respiratory distress syndrome (ARDS) [1,2]. Patient characteristics, including age, various comorbidities and host genetic factors, are associated with an increased risk for severe illness or death due to COVID-19 [3,4]. In addition, contextual factors and organizational issues, including the strain on hospital capacity, can negatively impact patient outcomes [5,6]. Moreover, the risk for severe outcomes has been evolving over time as successive COVID-19 waves have been triggered by emerging SARS-CoV-2 variants with varying epidemiological characteristics [7]. These new variants accumulate mutations impacting their transmissibility, immune escape, and/or virulence [8,9,10].

Continuous genomic surveillance is essential to detect, monitor, and assess emerging virus variants. Infection with a SARS-CoV-2 variant can be confirmed using genomic detection techniques, such as whole-genome sequencing (WGS), or can be suspected based on diagnostic screening nucleic acid amplification technique (NAAT)-based assays (e.g., S gene target failure (SGTF) assays) in a context of known circulating variants [11]. The surveillance of SARS-CoV-2 variants in Belgium is based on WGS of a representative national sample of the positive cases using a sentinel laboratory network (baseline genomic surveillance), and is complemented by targeted sequencing of additional priority samples (active genomic surveillance) and targeted molecular markers aiming to detect and monitor variants of concern (VOCs) [12,13,14]. 

Several SARS-CoV-2 variants have become dominant globally over the course of the pandemic. In Belgium, the “wild-type” SARS-CoV-2 variant was predominantly present from the beginning of the pandemic, followed by the emergence of the Alpha variant in December 2020 [14]. The risk of hospitalization, ICU admission, and mortality was found to be higher for COVID-19 patients with the Alpha variant compared to the “wild-type” variant, suggesting higher virulence of the Alpha variant [15,16,17,18,19]. The Belgian national vaccination campaign against COVID-19 started in early 2021, just before the onset of the third wave, during which, the Delta variant emerged. In the week of the 5th of July 2021, the Delta variant represented more than 80% of circulating strains [14]. The severity of illness was purported to be higher among Delta cases; however, studies comparing Alpha and Delta variants showed conflicting results [20,21,22,23]. On the 11th and 14th of November 2021, the Omicron SARS-CoV-2 strain was identified by the genomic surveillance system of Botswana and South Africa, respectively [24]. This variant carries a larger number of mutations than preceding variants [25]. Some mutations are described to lead to increased transmissibility and immune evasion compared to preceding variants [26,27,28,29,30]. Subsequently, Omicron was designated a VOC by WHO on the 26th of November 2021 [24]. On the same day, Belgium reported that this new variant was found in an unvaccinated international traveler returning from Egypt [31]. The growth advantage of the Omicron VOC over the Delta VOC subsequently led to a rapid spread in the community [32]. Though the booster vaccination campaign started in Belgium for the elderly in September 2021, the rapid rise of the highly infectious and immune-escaping Omicron variant during the end of 2021 pushed policy-makers to accelerate the administration of booster shots to the general population. By the first week of January 2022, the Omicron, B.A.1 and, B.A.1.1 variants jointly represented more than 80% of the positive samples among baseline genomic surveillance in Belgium [14]. Subsequently, the Omicron, B.A.2 variant emerged and became predominant, reaching 80% of circulating strains by the week of the 7th of March 2022 [14]. A number of studies in South Africa, the United Kingdom (UK), Norway, Canada, and the United States (US) have shown a reduced risk of clinical severity of Omicron infections as compared to Delta infections [33,34,35,36,37,38,39,40,41,42,43]. However, making inferences about Omicron’s intrinsic virulence as compared to preceding variants is challenging given the different levels of preexisting immunity in the affected populations and the oversaturation of hospital capacity during surges.

In the present study, we assess the clinical severity of the Omicron versus the Delta variant in the Belgian hospitalized COVID-19 population of the autumn-winter season of 2021–2022, by accounting for both patient characteristics, including demographics, underlying comorbidities, and preexisting natural and vaccine-induced immunity, as well as organizational hospital characteristics.

## 2. Materials and Methods

### 2.1. Causal Framework and Data Linkage

The study is conducted within a causal conceptual framework to assess the effect of SARS-CoV-2 variants on COVID-19 disease severity among hospitalized patients [12]. This framework describes the causal model and the data infrastructure (LINK-VACC project) which allows individual-level data linkage of selected variables from existing registries through the national registry number, including data on hospitalized COVID-19 patients [44], COVID-19 test results (including sequencing information) [45], administered COVID-19 vaccines, and socio-economic indicators.

### 2.2. Selection Study Population

The study population consisted of hospitalized symptomatic COVID-19 patients who were admitted to a Belgian hospital between 1 September 2021 and 28 March 2022, and for whom an admission form was reported in the Clinical Hospital Survey (CHS) [44]. The 1^st^ of September 2021 marks the start of the booster campaign in Belgium. Moreover, no Alpha variant (also causing SGTF on the TaqPath COVID-19 PCR test) has been detected in Belgium after this date. The data were extracted on 11 April 2022 to allow for at least two weeks of follow-up of recently admitted patients. The analysis was restricted to adults (≥18 years) with a laboratory-confirmed COVID-19 infection (RT-PCR and/or rapid antigen test) who were tested because of COVID-19 symptoms, and with available viral genetic/genomic data (see further). Patients that were transferred, readmitted, or hospitalized in a psychiatric hospital or in a hospital without an intensive care unit (ICU) were excluded from the study population, in order to avoid incomplete clinical episodes within the analysis. Further, patients with a missing date of diagnosis, date of admission, or date of discharge were excluded, as well as patients with a missing national registry number, which is required to link information from the different COVID-19 registries.

### 2.3. Exposure

Viral genetic or genomic information indicating infection with the Omicron or Delta variant was obtained through the linkage of the national registry number of hospitalized patients meeting the inclusion criteria with the COVID-19 TestResults Database. Both confirmed variant results, i.e., identified through WGS, and compatible variant results, e.g., identified through a probe PCR assay, were considered for the analysis. The variant results were obtained from both the baseline and active genomic surveillance. All Omicron sublineages were considered for the analysis. However, at the time of data extraction, only, B.A.1, BA.1.1, and, B.A.2 were circulating among hospitalized COVID-19 patients in Belgium. Therefore, no other sublineages were detected in hospitalized COVID-19 patients registered in the CHS. To assure that the hospital admission was related to the detected infection, patients with their sample collected more than 20 days before hospital admission or after hospital discharge were excluded from the study population. 

### 2.4. Matching

Patients in the Omicron group were matched to patients in the Delta group based on the hospital to which they were admitted using the *MatchIt* package (*matchit* function with *method = “exact”*) [46] in R. Stratification weights were calculated for the matched units. Patients were not matched based on other variables, as the matching process would result in discarding observations and smaller sample sizes. The other potential confounding variables, as identified in a causal model [12], were accounted for using a multivariable regression analysis (see further). 

### 2.5. Outcome

The clinical severity among the hospitalized study population was assessed by comparing the development of severe COVID-19 (defined as either ICU admission, ARDS development, or in-hospital mortality) between both exposure groups (Delta or Omicron). In addition, we specifically looked at the difference in risk for ICU admission and in-hospital mortality. 

### 2.6. Covariates

Vaccination status was defined as ‘not or partially vaccinated’ when cases diagnosed with COVID-19 had not received a dose of vaccine, had received only one dose of a two-dose schedule, or had not reached 14 days after the complete primary vaccination (two doses for BNT162b2, ChAdOx1, and mRNA-1273, and one dose for Ad26.COV2.S); ‘primary course completed’ when diagnosed ≥ 14 days after the complete primary vaccination; and ‘primary course completed plus booster’ when diagnosed ≥ 14 days after the booster dose. Reinfections were defined as two positive tests in the same individual taken more than 90 days apart [47]. Patients were categorized in three groups according to their underlying comorbidities associated with a moderate or high risk of severe COVID-19 (i.e., no underlying comorbidities, medium-risk comorbidity, and high-risk comorbidity) [21]. Medium-risk comorbidities include chronic liver disease, immunosuppressive therapy, diabetes, chronic lung disease, obesity, cognitive disorder, or cardiovascular disease. High-risk comorbidities include people with an organ transplant, immunodeficiency, hematological cancer, solid cancer, active hematological or solid cancers, chronic neurological disease, or chronic kidney disease. The ICU occupancy rate was defined as the number of recognized ICU beds occupied by COVID-19 patients averaged over the patient’s hospital stay. The place of infection was defined as ‘community-acquired’ when the symptom onset was up to and including 7 days after admission, ‘nursing home acquired’ when it concerned a nursing home resident and the symptom onset was up to and including 7 days after admission, and ‘hospital acquired’ when the symptom onset was 8 days or more after hospital admission [48]. The socio-economic indicators included were educational level at the individual level, income at the household level, and the population density and the median net taxable income per capita at the residential postal code level. Educational level was first classified using the International Standard Classification of Education (ISCED) [49]. These different categories were simplified into three education levels: low (ISCED0 to ISCED2, respectively, ranging from ‘less than primary education’ to ‘lower secondary school’), middle (ISCED3 to ISCED4, respectively, ranging from ‘upper secondary school’ to ‘post-secondary non tertiary education’), and high (ISCED5 to ISCED 8, respectively, ranging from ‘short-cycle tertiary education’ to ‘doctoral or equivalent level’). Income was computed from the net income of the household and divided in deciles: low income (decile 1 to 4), middle income (decile 5 to 7), high income (decile 8 to 10). 

### 2.7. Statistical Analyses

Demographic and clinical information of the matched study population was presented by variant. The categorical variables were described as frequency rates, and percentages and continuous variables were described using the median and interquartile range (IQR). Univariate comparisons of categorical variables were performed using χ^2^ or Fisher’s exact tests, and continuous variables were compared using the Wilcoxon–Mann–Whitney rank-sum test.

Twenty-fold multiple imputation of missing values was performed using the *mice* package [50] for all covariates used in the multivariable model (see further). Binary, categorical, and numerical variables were imputed with logistic regression, multinomial regression, and predictive mean matching, respectively. The imputation was performed using thirty iterations to achieve good convergence of the Markov chain Monte Carlo (MCMC), and the visit sequence was set from a low to high proportion of missing data. 

A weighted logistic regression model (using matching weights) was fitted with the following covariates (as identified in the causal model [12]): age, gender, comorbidity group, place of infection, educational level, income, population density at postal code level, vaccination status at diagnosis, mean ICU occupancy rate during the patient’s hospital stay, and two-way interactions of these covariates with the SARS-CoV-2 variant. Numeric variables were entered in the model with linear and quadratic terms. 

Regression standardization [51], which involves simulating the average potential outcomes under each variant, was performed to obtain the marginal effect with respect to the remaining matched sample. The causal effect was estimated as a contrast between those average potential outcomes in terms of a relative risk (RR) and a risk difference (RD). Block bootstrapping [52] of matched pairs (B = 1000 replications) was performed on each imputed dataset [53] to estimate the variance on each parameter of interest. Pooled point estimates and confidence intervals were then obtained using Rubin’s rules for multiple imputation [54].

All analyses were conducted in R 4.0.1 [55]. 

### 2.8. Sensitivity Analyses

Three sensitivity analyses were performed by conducting the same modeling procedure as described above, but applied to (1) a study population excluding patients with a documented previous infection, to (2) a study population only including patients with a confirmed Omicron or Delta infection (obtained through WGS from both baseline and active genomic surveillance), and to (3) a study population only including patients with their sample selected in the context of the representative baseline surveillance. 

## 3. Results

### 3.1. Basic Descriptive Characteristics of the Matched Study Population

As recorded on 11 April 2022, the CHS database contained a total of 109,233 case records of COVID-19 patients (Figure 1). After deduplication and exclusion of patients not meeting the inclusion criteria, a total of 9779 hospitalized COVID-19 patients admitted between 1 September 2021 and 28 March 2022 were selected, and 1069 (11%) of them had available exposure information. These were either identified as having a confirmed or compatible Omicron infection (n = 448), or as having a confirmed or compatible Delta infection (n = 621) upon linkage with the COVID-19 TestResult Database. A total of 445 Omicron patients could be matched to 509 Delta patients based on the hospital, and a total of 954 cases were thus included in the descriptive analysis. A flowchart for the study population selection is presented in Figure 1.

The median age of the matched study population (n = 954) was 74 years (IQR: 63–83), 59% (564/954) were men, and 10% (90/942) were nursing home residents. The patients stayed for a median of 9 days (IQR: 5–17) in the hospital, and the median ICU bed occupancy rate averaged over the patient’s hospital stay was 19% (IQR: 13–27). More than half of the patients (56%; 532/952) had at least one underlying comorbidity associated with a high risk for severe COVID-19 [34], whereas 16% (151/952) of the patients had no underlying comorbidities defined as increasing the risk for severe COVID-19. One fifth of the patients (20%; 194/954) were not or partially vaccinated, 43% (412/954) had completed the primary course, and 36% (348/954) also had received a booster dose. Only 1% (13/954) of the patients had a documented previous infection. The overall occurrence of severe COVID-19 (defined as either ICU admission, ARDS development, or in-hospital mortality), ICU admission, and in-hospital mortality was 24% (230/947), 14% (137/952), and 15% (140/944), respectively. A total of 76% (386/509) of Delta infections were confirmed through WGS, whereas this was the case for 56% (251/445) of Omicron infections. A minority (12%; 54/445) of infections in the Omicron group were caused by the, B.A.2 sublineage, whereas the majority (88%; 391/445) were infections caused by the, B.A.1 or, B.A.1.1 Omicron (sub)lineages. 

Table 1 shows the patient characteristics by variant. Patients infected with Omicron were older on average; were more often nursing home residents; and more frequently had cardiovascular disease, chronic neurological disease, chronic cognitive deficit, and chronic renal disease. In contrast, they were less frequently immunocompromised and obese. Further, as a consequence of the vaccination rollout over time, a significantly higher proportion of Omicron patients had completed the primary course and had received a booster (65%) as compared to Delta patients (12%). The ICU occupancy rate averaged over the patient’s hospital stay was lower for the Omicron patients as compared to the Delta patients (i.e., 17% and 25% of recognized ICU beds occupied by COVID-19 patients, respectively). The Omicron patients stayed for a median of 8 days in the hospital, whereas the median length of stay was 10 days for the Delta patients. 

### 3.2. Causal Inference Estimates

The estimated standardized risks (with respect to the model and the covariate distribution) for severe COVID-19, ICU admission, and/or in-hospital mortality among hospitalized patients are presented for both variants in Table 2. The estimated standardized risk for severe COVID-19 (defined based on a combination of severity indicators, including ARDS, ICU admission, and in-hospital mortality) in hospitalized patients was significantly lower when infected with the Omicron variant as compared to when infected with the Delta variant (RR = 0.63; 95% CI (0.30; 0.97)). When only looking at ICU admission as a severity indicator, the risk was also significantly lower when infected with the Omicron variant as compared to when infected with the Delta variant (RR = 0.56; 95% CI (0.14; 0.99)). There was no statistically significant difference in the risk of dying in the hospital between the two variants (RR = 0.78, 95% CI (0.28–1.29)).

### 3.3. Sensitivity Analyses

The occurrence of a previous infection was not included as a covariate in the logistic regression model given the low frequency (1%; 13/954) and the subsequent high probability of obtaining bootstrap resamples with zero counts for one of the exposure groups. Therefore, a sensitivity analysis assessed whether excluding patients with a documented previous infection from the study population would influence the results. The standardized risks and causal effect estimates within this subgroup (433 patients infected with the Omicron variant and 508 patients infected with the Delta variant) were similar as compared to the main analysis results (Appendix A).

A second sensitivity analysis was performed by only including the confirmed (through WGS) Omicron and Delta infections. The causal effect estimates within this subgroup (251 patients infected with the Omicron variant and 368 patients infected with the Delta variant) showed similar trends as those in the main analysis, albeit with higher standardized risks and wider confidence intervals (Appendix A).

A third sensitivity analysis was performed by only considering patients for whom the variant information was obtained through the representative baseline surveillance (i.e., their positive SARS-CoV-2 sample was selected *at random* by one of the sentinel laboratories). The causal effect estimates that were obtained within this subgroup (183 patients infected with the Omicron variant and 265 patients infected with the Delta variant) were no longer significant for any of the outcomes (Appendix A). However, similar trends were obtained for severe COVID-19 and ICU admission as in the main analysis, with a lower adjusted risk when infected with Omicron as compared to when infected with Delta. This in contrast to in-hospital mortality, for which an opposite trend was observed.

## 4. Discussion

This study applied a causal research methodology [12] to assess the severity of the Omicron compared to the Delta variant among hospitalized COVID-19 patients, using individually-linked data from routine COVID-19 surveillance systems in Belgium. The estimated risk for severe COVID-19 (defined as either ICU admission, and/or ARDS development, and/or in-hospital death) in hospitalized patients was significantly lower when infected with the Omicron variant as compared to when infected with the Delta variant (RR = 0.63; 95% CI (0.30; 0.97)) in an adjusted analysis accounting for the potential confounding variables as identified in a causal model [12] using a combination of matching and regression standardization. When looking specifically at ICU admission, the adjusted standardized risk was significantly lower when infected with Omicron (RR = 0.56; 95% CI (0.14; 0.99)). The present study confirms previous findings and contributes additional evidence that hospitalized patients infected with the Omicron variant are less likely to develop severe clinical outcomes. Indeed, one of the key findings of a study based on genome sequence analysis of 4468 Omicron samples conducted in Houston, Texas was that the maximum respiratory support required for Omicron patients was significantly less than that for Alpha or Delta patients [36]. Similarly, a nation-wide study in the US showed a marked decrease of the 3-day risk for mechanical ventilation in the Omicron cohort as compared to the Delta cohort [38]. In the absence of widespread availability of the patient’s virus genotyping data, two hospital-based studies in South Africa observed a decreased severity of disease (defined based on ICU admission, length of stay, requirement for oxygen treatment, or death) during the fourth wave (with Omicron dominance) compared with previous waves [41,56]. Another study in South Africa reported a 70% lower odds of severe disease (defined based on a combination of severity indicators including ICU admission, invasive ventilation, ARDS, or death) in hospitalized patients with a SGTF infection (used as a proxy for Omicron, B.A.1 infections) as compared to Delta variant infections [35]. 

A matched population-based study of more than 9000 Omicron cases in Ontario showed a significant decrease in the risk of death in Omicron cases as compared to Delta cases [33]. In our study population, a similar trend was observed with a lower crude proportion of in-hospital mortality among Omicron patients as compared to Delta patients. However, the adjusted analysis did not show a significant difference in the standardized risk for in-hospital mortality risk according to the variant (RR = 0.78, 95% CI (0.28–1.29)). Given the large confidence intervals, this may be due to the limited sensitivity of the study to detect an effect size of this magnitude. On the other hand, the identified associations within the hospitalized population may be different from those existing in the general population (i.e., lack of external validity) [57]. Moreover, restricting the analysis to hospitalized patients may induce collider bias, as the relationships between any risk factors that relate to hospitalization can be distorted compared to those among the general population [57].

The patients with an Omicron infection differed substantially from the patients with a Delta infection in terms of demographics and comorbidities, whereas no difference in socio-economic determinants of health was observed between both variant groups. The median age was higher in the Omicron group (78 year) than in the Delta group (71 year). There were more nursing home residents among the Omicron patients than the Delta patients. One of the possible factors could be the vaccination campaign in Belgium that targeted nursing home residents as the first group to receive the anti-SARS-CoV-2 vaccines in January 2021, followed by a booster campaign in September 2021. The first signs of a waning effect of the booster may have coincided with the emergence of the Omicron variant in November 2021. Another possibility is the immunosenescence in this frail patient group [58], which could lead to increased susceptibility to the more transmittable, less severe, and immune-escaping Omicron variant. Moreover, the transmission in nursing homes may have been enhanced by relaxing the mitigation measures during the Omicron wave. Remarkably, Omicron patients appear to be younger as compared to Delta patients in other studies studying hospitalized patients [35,36,41,56]. Differences can be multifactorial, such as approach in and timing of the vaccination campaign, testing strategy, or social contact patterns within or across age groups. The other observed discrepancies in patient characteristics between Omicron and Delta patients are often interlinked (e.g., lower prevalence of immunocompromising disorders and obesity among the nursing home population) and/or could be potentially explained by multiple factors, including the social behavior or mobility in the respective patient groups. Nevertheless, the observed differences in characteristics between Omicron and Delta patients were adjusted for in the multivariable analysis.

A number of limitations need to be considered. First, it is important to acknowledge the study selection procedures when interpreting the risk estimates. Restricting the analysis to hospitalized patients may hamper both external and internal validity [57,59,60]. Further, the hospitalized patients with available information on the SARS-CoV-2 variant of their infection may not be representative of the hospitalized population. Selection bias may arise, as samples are selected for sequencing based on the viral load and/or based on certain indications in the context of active genomic surveillance, such as breakthrough infections. However, these indications were constant over time during both the Delta and the Omicron period [13]. A sensitivity analysis only considering samples selected in the context of baseline genomic surveillance revealed similar, albeit non-significant, trends as for the main analysis, except for in-hospital mortality. However, the small sample size resulted in large confidence intervals. Finally, the matched design should be taken into account when interpreting the standardized risks that are obtained after g-computation, whereby reference is made to the observed study population and for which the interpretation is tied to the particular population across which the marginalization was done. Following the matching of the Omicron patients to the Delta patients based on the hospital to which they were admitted, the obtained effect estimates correspond to the average effect in the remaining matched sample. 

A general limitation when assessing the severity of consecutive SARS-CoV-2 variants circulating during different time periods is the fact that the exposure groups differ considerably in terms of patient demography and contextual factors, such as vaccination rollout, testing strategy, and dynamics of the respective waves. Having exhaustive information on the vaccination status, and performing a sensitivity analysis excluding patients with a documented previous infection, enabled us to account, to a large extent, for the gradually changing background immunity in the population. However, the number of documented reinfections, defined as two positive tests in the same individual taken more than 90 days apart, is probably an underestimation given the larger antigenic diversity of Omicron [59]. The effects of undocumented previous infections and the fact that the rate of underascertainment varies considerably over time could potentially hamper the comparison. Next to the changing immunity status, severity analyses could also be affected by the changes in testing strategy. Indeed, the near universal, pre-hospitalization COVID-19 testing and the increasing population prevalence of infection during the Omicron wave impacts the possibility of some patients being hospitalized with, rather than for, SARS-CoV-2 infection [60]. Therefore, we have excluded hospitalized patients that tested positive for COVID-19 in a screening context (i.e., without the development of COVID-19 symptoms) in the current analysis. Although we were able to adjust for a large number of known confounding variables, residual confounding may exist when the circulation of the Delta or Omicron variants coincides with important time-varying factors for which we are unable to adjust due to the lack of data (e.g., co-infections with other respiratory pathogens), or of which we are currently unaware.

A third limitation is the potential measurement bias related to the inclusion of compatible Omicron and Delta cases in the main analysis. Indeed, multiplex mutation-specific PCR-based assays are not always able to definitively distinguish different SARS-CoV-2 variants. Therefore, a sensitivity analysis only including the confirmed Omicron and Delta infections through WGS was conducted. This analysis revealed similar conclusions (lower standardized risks for the severe outcomes when infected with Omicron as compared to when infected with Delta), albeit with different values for the standardized risks, potentially due to a selection bias when considering samples for WGS.

Fourth, the, B.A.2 sublineage reached more than 80% in the baseline surveillance during the week of 22th of March 2022 [14]. Still, the fraction of, B.A.2 infections in the current (hospitalized) study population is relatively low. Studies have shown that the infectiousness and the evading capability of neutralizing antibodies of the, B.A.2 sublineage are higher than the other BA sublineages [61,62]. Future analyses should provide more insights regarding the clinical severity of the, B.A.2 sublineage compared to, B.A.1 [63]. 

Lastly, conducting the analysis during the fifth wave dominated by Omicron might have led to an overrepresentation of recently admitted Omicron patients with a short hospital stay. We partly accounted for this limitation by applying a data cutoff of two weeks to ensure sufficient follow-up time. 

The key strengths of this study are: (1) the availability of detailed information on the characteristics and outcomes of hospitalized patients; (2) the linkage with socio-economic data, sequencing results, and the exhaustive vaccination register; and (3) the multi-center nature of the study providing a nation-wide coverage, increasing the generalizability. Another important strength of the study design is the matching of patients based on the hospital to which they were admitted: this accounts for between-center variability in patient outcomes and differences that may exist in the decision-making process to admit (severe) COVID-19 patients and possibly in treatment protocols [6,64]. Moreover, linking data from the exhaustive Surge Capacity Survey, which serves as an operational survey during the Belgian SARS-CoV-2 pandemic regarding bed occupancy in hospitals, enabled us to account, in our model, for the hospital-specific organizational characteristics such as the overflow of recognized ICU beds, which has previously been reported to be associated with a higher odds of in-hospital mortality [6], and which may be associated in time and place with the circulation of a particular SARS-CoV-2 variant. Other important confounders, identified when establishing a causal model, were adjusted for in the multivariable logistic regression analysis. 

## 5. Conclusions

We observed a lower risk for severe COVID-19 (based on a combination of severity indicators) and ICU admission in Belgian hospitalized COVID-19 patients when infected with the Omicron variant as compared to when infected with the Delta variant. Inferring the intrinsic severity of SARS-CoV-2 variants is challenging and requires detailed information on potential confounders as identified in a causal model. This study demonstrates the added value of genomic surveillance during the COVID-19 pandemic and the subsequent individual-level linkage with other surveillance registries to better understand the complex and multifactorial nature of COVID-19 disease severity, and to deliver evidence-based results to policy-makers.

## Figures and Tables

**Figure 1 viruses-14-01297-f001:**
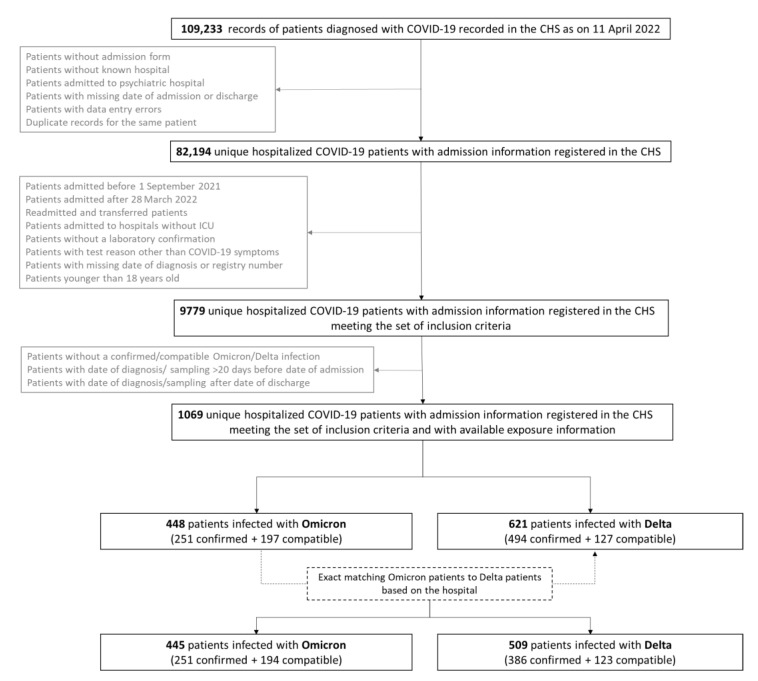
Flow chart of the study population selection within a multi-center matched cohort study to assess the impact of SARS-CoV-2 variants on COVID-19 disease severity among hospitalized patients admitted between 1 September 2021 and the 28 March 2022 in Belgium.

**Table 1 viruses-14-01297-t001:** Patient characteristics by variant (Omicron and Delta) within a multi-center matched cohort study to assess the impact of SARS-CoV-2 variants on COVID-19 disease severity among hospitalized patients admitted between 1 September 2021 and the 28 March 2022 in Belgium.

	Patients Infected with Omicron (n = 445)			Patients Infected with Delta (n = 509)			
	n	%	N	n	%	N	*p* Value
**Demographics**							
Age (years), median (IQR)	78 (67–86)		445	71 (58–80)		509	<0.001
Male gender, n (%)	277	62.2	445	287	56.4	509	0.077
Nursing home resident, n (%)	68	15.6	436	22	4.3	506	<0.001
**Comorbidities**							
Cardiovascular Disease, n (%)	228	51.4	444	185	36.4	508	<0.001
History of Arterial Hypertension, n (%)	197	44.4	444	205	40.4	508	0.24
Diabetes mellitus, n (%)	125	28.2	444	136	26.8	508	0.69
Obesity, n (%)	48	10.8	444	102	20.1	508	<0.001
Chronic Pulmonary Disease, n (%)	107	24.1	444	119	23.4	508	0.87
Chronic Neurological Disease, n (%)	93	20.9	444	62	12.2	508	<0.001
Chronic Cognitive Deficit, n (%)	68	15.3	444	33	6.5	508	<0.001
Chronic Renal Disease, n (%)	143	32.2	444	106	20.9	508	<0.001
Chronic Liver Disease, n (%)	14	3.2	444	12	2.4	508	0.55
Solid Cancer, n (%)	97	21.8	444	87	17.1	508	0.079
Hematological Cancer, n (%)	25	5.6	444	28	5.5	508	>0.99
Solid organ transplantation, n (%)	11	2.5	444	27	5.3	508	0.030
Chronic Immunosuppression, n (%)	21	4.7	444	55	10.8	508	0.001
Comorbidity groups ^1^, n (%)							<0.001
No underlying comorbidities	53	11.9	444	98	19.3	508	-
Medium-risk comorbidity	113	25.5	444	156	30.7	508	-
High-risk comorbidity	278	62.6	444	254	50.0	508	-
**Socio-economic status**							
Education level ^2^, n (%)							0.59
Low	219	62.8	349	227	60.1	378	-
Middle	80	22.9	349	99	26.2	378	-
High	50	14.3	349	52	13.8	378	-
Income (decile) ^3^, n (%)							0.20
Low income	212	54.9	386	219	49.7	441	-
Middle income	120	31.1	386	143	32.4	441	-
High income	54	14.0	386	79	17.9	441	-
Population density ^4^, median (IQR)	990 (470–2600)		426	720 (380–1700)		503	<0.001
Median taxable income per capita ^5^, median (IQR)	27,000 (24,000–29,000)		426	28,000 (26,000–29,000)		503	0.15
**Place of infection**, **n (%)**							<0.001
Community-acquired	334	76.4	437	459	90.9	505	-
Hospital-acquired ^6^	38	8.7	437	25	5.0	505	-
Nursing home-acquired	65	14.8	437	21	4.2	505	-
**Preexisting immunity**							
Documented previous infection, n (%)	12	2.7	445	1	0.2	509	0.001
Vaccination status ^7^, n (%)							<0.001
Not or partially vaccinated	66	14.8	445	128	25.1	509	-
Primary course completed	90	20.2	445	322	63.3	509	-
Primary course completed and booster	289	64.9	445	59	11.6	509	-
**Hospital organizational characteristics**							
ICU occupancy rate ^8^, median (IQR)	17 (11–20)		445	25 (15–33)		509	<0.001
**Disease characteristics**							
CRP ^9^ (mg/l) on admission, median (IQR)	35 (14–72)		428	66 (29–130)		503	<0.001
**Clinical outcomes**							
Severe ^10^ COVID-19, n (%)	69	15.7	440	161	31.8	507	<0.001
ICU admission, n (%)	31	7.0	445	106	20.9	507	<0.001
In-hospital mortality, n (%)	51	11.6	438	89	17.6	505	0.012
Invasive ventilation, n (%)	8	1.8	440	40	8.1	494	0.37
ECLS ^11^, n (%)	2	0.5	440	2	0.4	494	0.21
ARDS ^12^, n (%)	12	2.7	445	60	11.8	509	<0.001
Hospital length of stay (days), median (IQR)	8 (5–15)		445	10 (5–18)		509	0.001
ICU length of stay (days), median (IQR)	5 (2–7)		26	9 (5–17)		92	0.003

^1^ Comorbidity groups classified based on patient’s underlying comorbidities associated with a moderate or high risk of severe COVID-19. Medium risk includes people with chronic liver disease, immunosuppressive therapy, diabetes, chronic lung disease, obesity, cognitive disorder, and cardiovascular disease. High risk includes people with a solid organ transplant, immunodeficiency, hematological cancer, other active cancers, chronic neurological disease, and chronic kidney disease. ^2^ Low: ISCED0 to ISCED2; Middle: ISCED3 to ISCED4; and High: ISCED5 to ISCED 8. ^3^ Low income: decile 1 to 4; Middle income: decile 5 to 7; and High income: decile 8 to 10. ^4^ Population density at the postal code level of the residence of the patient. ^5^ Median net taxable income per capita at the postal code level of the residence of the patient. ^6^ Symptom onset or diagnosis more than 8 days after hospital admission. ^7^ Not or partially vaccinated: diagnosed when no dose received, one dose out of a two-dose schedule, or before 14 days after the primary vaccination schedule; Primary course completed: diagnosed ≥ 14 days after the primary vaccination schedule (2 doses for BNT162b2, ChAdOx1, and mRNA-1273, and one dose for Ad26.COV2.S); Primary course completed and booster: diagnosed ≥ 14 days after the booster dose. ^8^ Defined as the number of recognized ICU beds occupied by COVID-19 patients averaged over the patient’s hospital stay. ^9^ C-reactive protein. ^10^ Defined as a combination of three binary severity indicators: having been admitted to ICU and/or developed acute respiratory distress syndrome (ARDS) and/or died in the hospital. ^11^ Extracorporeal life support. ^12^ Acute respiratory distress syndrome.

**Table 2 viruses-14-01297-t002:** Standardized risk by variant (in %), relative risk (RR), and risk difference (RD, in %) estimates, and 95% confidence interval (CI) for severity outcomes within a multi-center cohort study to assess the impact of SARS-CoV-2 variants on COVID-19 disease severity among hospitalized patients in Belgium.

Outcome	Standardized Risk [95% CI] in %		RR [95% CI]	RD [95% CI] in %
	Omicron	Delta		
Severe COVID-19 ^1^	22.3 [12.0; 32.6]	35.2 [26.5; 43.8]	0.63 [0.30; 0.97]	−12.9 [−26.7; 0.1]
ICU admission	12.2 [3.0; 21.4]	21.7 [17.7; 25.8]	0.56 [0.14; 0.99]	−9.5 [−19.1; 0.1]
In-hospital mortality	19.1 [9.2; 29.0]	24.4 [16.1; 32.6]	0.78 [0.28; 1.29]	−5.2 [−17.8; 7.2]

^1^ Defined as either intensive care unit (ICU) admission, and/or acute respiratory distress syndrome (ARDS), and/or in-hospital mortality.

## Data Availability

The individual-level datasets generated or analyzed during the current study do not fulfill the requirements for open data access. The data is too dense and comprehensive to preserve patient privacy. The data of the individual data sources (Clinical Hospital Survey, Vaccinnet+, COVID-19 TestResult Database, and StatBel) within the LINK-VACC project are kept in the pseudonymized environment of healthdata.be, and a link between the individual data in each of them takes place thanks to the use of a pseudonymized national reference number managed by healthdata.be under a ‘project mandate’. A ‘project mandate’ consists of a group of individuals, a group of variables, and a time period. Access rights to the pseudonymized data in the healthdata.be data warehouse are granted ad nominatum for the scientists involved in the surveillance activities at Sciensano. External investigators with a request for selected data should fill in the data request form (https://epistat.wiv-isp.be/datarequest, accessed on 13 June 2022). Depending on the type of desired data (anonymous or pseudonymized), the provision of data will have to be assessed by the Belgian Information Security Committee Social Security & Health based on legal and ethical regulations, and is outlined in a data transfer agreement with the data owner (Sciensano).

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
