# Peer review of "Clinical Severity of SARS-CoV-2 Omicron Variant Compared with Delta among Hospitalized COVID-19 Patients in Belgium during Autumn and Winter Season 2021–2022"

_viruses, 2022, doi:10.3390/v14061297_

Round 1

Reviewer 1 Report

VanGoethem et.al have performed a good epidemiological analysis that helps us to understand the difference in severity between Omicron and delta variants. Despite the limitations the study does support that Omicron is a less severe variant although vaccination programmes have played a major part in reduction of severe COVID-19 illness

Methods

Page 3:  section 2.4 Matching:

  1. It is unclear if propensity scores were generated for matching? If so, this must be made clear.
  2. I can see from figure 1 that exact matching was used. What were the co-variates included in matching? This must be made clear in this section.

Results

  1. Table 1: The table should include a separate column to demonstrate the p-Value to understand the statistical difference in the co-variants between the two groups. e.g., median age 78 vs 71 (Is there a statistical difference?). Based on P-values the next question will arise why there remains a big difference between the groups in age and other co-variates despite matching?

  1. Table 2: Again, any reader would be interested to see a p-Value for the relative risks to clearly understand the statement “significant difference”. P-values to be included in text of manuscript as well.

  1. The legends used for Table-1 and Supplementary table S1 appears the same. As described in the text the legend must be made clear to understand the difference.

  1. Supplementary Table 2: This shows there is no difference between the groups even in severe COVID-19 infection RR-0.66 (0.29;1.03), A p-value would make this inference clear (importance of p-Value). This aspect is not highlighted in the manuscript text.

  1. Again, the legend of supplementary table 2 should be clearer by including the total numbers in each group so the reader can understand the table without looking at the manuscript text. This would apply for supplementary table 3 as well.

Discussion /limitations:

  1. Clearly a greater proportion have had their primary vaccination by the omicron period (64.9 vs 11.6). How is the vaccination component inputted in the risk analysis?  A subgroup analysis comparing those who completed their primary course of vaccination 379 vs 374 could help to delineate the difference between the influence of Vaccine vs Variant. (can be included as a Supplementary table if the authors feel this appropriate).

Reviewer 2 Report

Summary/General Reviewer Comments:  

The manuscript presents results of a retrospective multi-center matched cohort study of the clinical severity of children hospitalized with COVID while infected with either Omicron or Delta virus variants.  Patients were matched by hospital while possible confounding variables adjusted by using multivariable logistic regression analysis. Risk of severe COVID and ICU admission was significantly lower for children infected with Omicron.  This is a well-designed and executed study that confirms the lower virulence of the Omicron variant.

Specific (Minor) Reviewer Comments:

1.      Line 55.  The term “pandemic” instead of epidemic may be more appropriate here and elsewhere in the manuscript.

2.      Line 59.  The term “virulence” refers to the degree to which a pathogen can cause disease, whereas “higher severity” refers more to the disease itself.

3.      Line 83. “virulence” may be more appropriate than “severity” here.

4.      Line 145.  Was vaccine heterogeneity among study participants taken into account when comparing virus strain groups?

5.      Line 227 & Table 1.  A major potential confounder to this comparison would be differences in immunity between by two groups by vaccination or natural infection.  It is interesting that those with Omicron infections had 10x the number of documented previous natural infections and nearly 6x the number of completed/boosted vaccinations as those with Delta infections.  Undocumented natural infections are also likely present among the study participants.  Preexisting antibody could influence clinical outcomes.  Although the study results are probably driven by differences in viral strain virulence, how can the authors be certain without serologic evaluation of the study participants of presence and titer of preexisting anti-SARS-CoV-2 antibodies?

6.      Line 235 & Table 1.  Are patient co-infections with other respiratory pathogens taken into account when comparing SARS-CoV-2 virus strains?

Round 2

Reviewer 1 Report

Response to Reviewer 1 Comments

Point 1: Methods (Page 3: section 2.4 Matching): It is unclear if propensity scores were generated for matching? If so, this must be made clear.

Response 1: We performed exact matching of Omicron patients to Delta patients based on the hospital in which they were admitted. We used exact matching (based on the hospital), which is a form of stratum matching that involves creating subclasses based on unique combinations of covariate values and assigning each unit into their corresponding subclass so that only units with identical covariate values are placed into the same subclass. As such, the matching itself is not based on propensity scores. As we performed k:1 matching where not all exposed units receive k matches, weights are required to account for the differential weight of the matched unexposed units. Stratification weights are computed for the matched units. Weights are 1 for the “exposed” (here: Omicron patients) and p/(1-p) for the “unexposed” (here: Delta patients), where p is the stratum “propensity score”. The matching weights are subsequently used to estimate the marginal effect (i.e., by using a weighted logistic regression model that incorporates the matching weights). We have added the following to section 2.4 (L131-133): “Patients in the Omicron group were matched to patients in the Delta group based on the hospital to which they were admitted using the MatchIt package (matchit function with method = “exact”) in R. Stratification weights were computed for the matched units.”

Comments: Thank you. It is clearer now

Point 2: Methods (Page 3: section 2.4 Matching): I can see from figure 1 that exact matching was used. What were the co-variates included in matching? This must be made clear in this section.

Response 2: Both in section 2.4 and in Figure 1, it is stated that the Omicron patients were matched to the Delta patients based on the hospital only, to account for differences in levels of care and admission criteria. Hospital is the only variable that was used for the matching. We have added the following clarification in section 2.4 (L134-137): “Patients were not matched based on other variables as the matching process would result in discarding observations and smaller sample sizes. The other potential confounding variables, as identified in a causal model, were accounted for using a multivariable regression analysis (see further).”

Comments: Thank you. It is clearer now

Point 3: Results, Table 1: The table should include a separate column to demonstrate the p-Value to understand the statistical difference in the co-variants between the two groups. e.g., median age 78 vs 71 (Is there a statistical difference?). Based on P-values the next question will arise why there remains a big difference between the groups in age and other co-variates despite matching?

Response 3: The matching was only performed based on the hospital. As such, when using the matching weights, the different hospitals are perfectly balanced between both exposure groups. The other co-variates were adjusted for, during the analysis, using a multivariable logistic regression analysis. Table 1 presents the demographical and clinical information of the matched study population (i.e., matched on hospital) by variant. Consequently, the other co-variates are not balanced between both groups in Table 1.

Table 1 was intended to provide the reader with a description of the characteristics of the study population and to show whether the two exposure groups are similar or different, and to what extent. There was no intention to perform statistical inference on these demographic and clinical variables, as the decision to include co-variates in the subsequent multivariable model did not depend on potential statistically significant differences, but instead on the identification through our predefined causal model. Indeed, this study was conducted within a causal research framework [1], where the confounders as identified in the Directed Acyclic Graph (DAG) were adjusted for in the subsequent analysis. Following the causal research methodology, we restrict ourselves to only interpret the effect estimate of interest by carefully building a causal model. As a consequence, the multivariable regression was not performed to discuss coefficients of all the co-variates [2], but rather to obtain coefficients to be able to appropriately estimate the risk of severe outcome for the study population by variant.

Comments: Thank you for your detailed justification on using multivariate models and not showing p-values in table-1. If this is Journal of statistics or epidemiology where the readers are expertise in statistical methods, then clearly there is no need for include p-values. But for virus journal which is predominantly clinician read journal, one would expect to understand if there were any difference in the characteristics of the cohorts using a p-Value. I personally feel there is no value in comparing the characteristics of the two cohorts without using p- values although your intention is to compare this as you have stated above (Table 1 was intended to provide the reader with a description of the characteristics of the study population and to show whether the two exposure groups are similar or different, and to what extent.).

 The authors may disagree with this comment and I will leave it to the decision of Editors on this who may want to get a review from an expert statistician.

 Point 4: Results, Table 2: Again, any reader would be interested to see a p-Value for the relative risks to clearly understand the statement “significant difference”. P-values to be included in text of manuscript as well.

 Response 4: The causal effect estimates presented in Table 2 were obtained after fitting a weighted logistic regression model (adjusted for the confounders as identified in our causal model) and subsequent regression standardization to simulate the marginal effect under each exposure status (here: variant). Bootstrapping [3] was used to estimate the variance on each parameter of interest, and Rubin’s rule was used to pool the point estimates and confidence intervals of the multiply imputed data. We have chosen to present the confidence intervals of the risk ratios as they provide information about statistical significance, as well as the direction and strength of the effect [2,4–7].

 Comments: Thank you for your detailed justification. My comment is much similar as the earlier comment. If this is a Journal of statistics or epidemiology where the readers are expertise in statistical methods, then clearly there is no need for include p-values. But for virus journal which is predominantly clinician read journal, the understanding would be better using a p-Value.

The authors may disagree with this comment and I will leave it to the decision of Editors on this who may want to get a review from an expert statistician.

Point 5: Results: the legends used for Table-1 and Supplementary table S1 appears the same. As described in the text the legend must be made clear to understand the difference.

Response 5: We have harmonized the definition of “severe COVID-19” in Tables 1 and 2, and Supplementary table S1, S2, and S3. For Supplementary table S1, the title states that it concerns the study population without a documented previous infection, while both those with and without previous documented infections are included in the main analysis. For Supplementary table S2, the title states that it concerns the study population with a confirmed (through whole genome sequencing) Omicron or Delta infection, while the main analysis includes both confirmed and suscpected (through presumptive genotyping methods) Omicron or Delta infections. For Supplementary table S3, the title states that it concerns the study population when only considering variant information obtained through the genomic baseline surveillance, while the main analysis includes variant information obtained through both baseline and active genomic surveillance (i.e., irrespective of the indication for sequencing).

Comments: Thank you. It is clearer now

Point 6: Results, Supplementary Table 2: This shows there is no difference between the groups even in severe COVID-19 infection RR-0.66 (0.29;1.03), A p-value would make this inference clear (importance of p-Value). This aspect is not highlighted in the manuscript text.

Response 6: As explained in Response 4, we have chosen to present the confidence intervals of the risk ratios as they provide information about statistical significance (based on whether the null value – 1 for RR and 0 for RD – lies within the limits of the interval), as well as the magnitude and precision of the estimated effect.

Comments: Thank you for your detailed justification. My comment is much similar as the earlier comment for response 4. If this is a Journal of statistics or epidemiology where the readers are expertise in statistical methods, then clearly there is no need for include p-values. But for virus journal which is predominantly clinician read journal, the understanding would be better using a p-Value.

The authors may disagree with this comment and I will leave it to the decision of Editors on this who may want to get a review from an expert statistician.

Point 7: Results: Again, the legend of supplementary table 2 should be clearer by including the total numbers in each group so the reader can understand the table without looking at the manuscript text. This would apply for supplementary table 3 as well.

Response 7: We agree with the reviewer and have added the numbers in the legend of Supplementary tables S1, S2, and S3.

Comments: Thank you. It is clearer now

Point 8: Discussion/limitations: clearly a greater proportion have had their primary vaccination by the omicron period (64.9 vs 11.6). How is the vaccination component inputted in the risk analysis?  A subgroup analysis comparing those who completed their primary course of vaccination 379 vs 374 could help to delineate the difference between the influence of Vaccine vs Variant. (can be included as a Supplementary table if the authors feel this appropriate).

Response 8: In this study, we have limited ourselves to investigate the causal effect of the SARS-CoV-2 variant on disease severity by applying a causal research methodology and building a causal model specific to the exposure-outcome relationship of interest. Our causal model assumes that vaccination status may confound this relationship. Vaccination status was defined as ‘not or partially vaccinated’ when cases diagnosed with COVID-19 had not received a dose of vaccine, had received only one dose of a two-dose schedule or had not reached 14 days after the complete primary vaccination (two doses for BNT162b2, ChAdOx1 and mRNA-1273 and one dose for Ad26.COV2.S); ‘primary course completed’ when diagnosed ≥14 days after the complete primary vaccination; and ‘primary course completed plus booster’ when diagnosed ≥14 days after the booster dose. To eliminate confounding from the association of interest, vaccination status was included as a co-variate (as well as its interaction effect with the variant) in a multivariable logistic regression model. As such, the analyses are adjusted for the vaccination status. The impact of vaccination status on COVID-19 disease severity, and how this differs depending on the SARS-CoV-2 variant of the infection, is indeed an interesting research question and will be subject to a different study.

Comments: Thanks

Reviewer 2 Report

Responses to reviewer comments are acceptable.

Author Response

No additional comments were provided.